# ID-Booth: Identity-consistent image generation with diffusion models

## Abstract

The recent retraction of large-scale biometric datasets, prompted by strict privacy regulations, presents a critical challenge for future biometric research. This is evident with the face recognition task, for which large-scale datasets were often gathered through web-scraping without the consent of subjects. A potential solution entails the creation of synthetic data, suitable for training recognition models, with deep generative models. Existing generative approaches rely on conditioning and fine-tuning of powerful pretrained diffusion models to achieve the synthesis of realistic images of a desired identity. Yet, these methods often do not consider the identity of subjects during training, leading to poor consistency between generated and intended identities. In contrast, methods that employ identity-based training objectives tend to overfit on various aspects of the identity, and in turn, lower the diversity of images that can be generated. To address these issues, we present the ID-Booth fine-tuning framework, which utilizes a novel triplet identity training objective and enables identity-consistent image generation while retaining the synthesis capabilities of pretrained models. Experiments across two latent diffusion models with varying prompt complexity reveal that our method facilitates better intra-identity consistency and inter-identity separability while achieving higher image diversity. In turn, the produced data enables the training of better-performing recognition models than even real-world datasets of a similar scale gathered with suitable consent. The source code for the ID-Booth framework is available at `omitted_for_review`.

## 1 Introduction

Deep neural networks are nowadays utilized as backbones in a variety of recognition systems (Bai et al., 2021). To achieve state-of-the-art performance these models require diverse large-scale training datasets, which are commonly gathered through web-scraping. However, this process presents an inadequate solution in the field of image-based biometrics, where strict regulations accompany the collection, distribution and use of data without the proper consent of subjects (Jasserand, 2018; Meden et al., 2021). This is especially evident when considering the face modality, for which several recognition datasets have already been retracted since the introduction of recent privacy acts and data-protection legislation, e.g. the GDPR (Hoofnagle et al., 2019). Meanwhile, manually gathering suitable datasets with the proper consent of subjects presents a time consuming process and often results in small-scale datasets captured in a constrained setting with limited diversity.

To ensure the future development of face recognition systems, researchers have proposed to instead rely on synthetic data for training (Boutros et al., 2023c). Nowadays diverse datasets of high-quality synthetic images can be generated with deep generative models, which have experienced rapid development in the past decade (Karras et al., 2019; Ho et al., 2020). Diffusion models currently represent the state-of-the-art among image generation models, as they offer unparalleled synthesis capabilities in terms of quality and diversity, while enabling synthesis guided by text prompts (Rombach et al., 2022). Recently, diffusion models have also been utilized to produce biometric datasets suitable for recognition tasks, i.e. containing images of multiple identities with multiple samples each. To this end, approaches rely on identity-conditioning (Ye et al., 2023; Papantoniou et al., 2024; Wang et al., 2024) and fine-tuning (Ruiz et al., 2023; Peng et al., 2024) of pretrained models. Nevertheless, most solutions focus mainly on image reconstruction during training, resulting in poor consistency between desired identities and generated ones. To address this issue, PortraitBooth (Peng et al.,

Real                                    ID-Booth samples

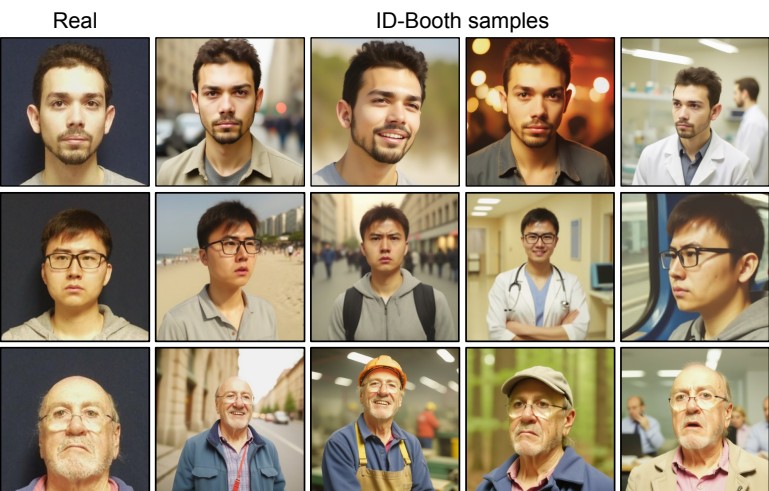

Figure 1: **Samples generated with the proposed ID-Booth framework.** The framework enables fine-tuning of pretrained diffusion models for generating diverse identity-consistent images based on images gathered in a constrained setting with the consent of subjects.

2024) recently extended the fine-tuning DreamBooth (Ruiz et al., 2023) method with an identity-based training objective. However, the proposed solution only considers the similarity between real samples of the desired identity and the generated samples during training. In turn, it tends to overfit on input identities, resulting in lower diversity of generated images.

In this paper, we present a solution for the outlined issues, in the form of a new fine-tuning framework, called ID-Booth. The proposed framework utilizes a novel triplet identity objective, which considers both positive and negative identity samples during training, to facilitate the generation of identity-consistent images while retaining the synthesis capabilities of pretrained models. Throughout the experiments, we explore the suitability of ID-Booth for addressing privacy concerns by generating diverse synthetic in-the-wild images of identities from the Tufts Face Database (Panetta et al., 2018), which contains images gathered in a constrained setting with the consent of subjects, as shown in Figure 1. We perform fine-tuning of two state-of-the-art diffusion models conditioned on text prompts of varying complexity and compare synthesis results with DreamBooth (Ruiz et al., 2023) and PortraitBooth (Peng et al., 2024) in terms of image quality, fidelity and diversity as well as intra-identity consistency and inter-identity separability. Furthermore, we investigate the utility of synthetic data by utilizing produced datasets to train face recognition models and evaluating their performance on five real-world verification benchmarks. We demonstrate that our fine-tuning framework enables the generation of datasets with better intra-identity consistency and inter-identity separability, both among synthetic samples or between synthetic and real ones. Consequently, this results in the training of more powerful recognition models than even with real-world datasets of a similar scale. Overall, the paper makes the following contributions:

- We introduce ID-Booth, a new fine-tuning framework for generating highly-diverse identity-consistent privacy-preserving images based on training images captured in a constrained setting with the consent of subjects.
- We propose a novel triplet identity training objective that improves identity consistency while better retaining the diversity and text-based control of pretrained diffusion models.
- We demonstrate the suitability of the produced datasets for training recognition models that outperform those trained on similar-scale real-world datasets gathered with consent.

## 2 RELATED WORK

**Image generation.** The field of image synthesis has undergone rapid development since the introduction of deep generative models. Generative Adversarial Networks (GANs) (Goodfellow et al., 2014) were the initial models to achieve the synthesis of convincing images, with a generator and a

discriminator network. Extensive improvements followed, namely StyleGAN (Karras et al., 2019) facilitated higher image quality and better control over the generation process. However, the synthesis capabilities of GANs have nowadays been surpassed by recent diffusion models (Dhariwal & Nichol, 2021), which generate images by gradually removing noise from initial noisy samples. This denoising process is learned with a convolutional encoder-decoder by predicting the noise that is added to training samples at different scales (Ho et al., 2020). Recently, Latent Diffusion Models (LDMs) (Rombach et al., 2022) achieved improved efficiency and efficacy by moving the denoising process from the pixel space to a lower-dimensionality latent space of a pretrained variational autoencoder. Their remarkable synthesis capabilities and conditioning on text prompts via a pretrained text encoder have led to their broad adoption, namely of the open-source Stable Diffusion model (Rombach et al., 2022). Image resolution has also been further improved with Stable Diffusion XL (SD-XL) (Podell et al., 2024), which utilizes a larger U-Net backbone along with two text encoders and additional conditioning schemes. Recent approaches have further enhanced control over the generation process, e.g. ControlNet (Zhang et al., 2023) conditions the model on segmentation masks or depth maps via an auxiliary trainable copy of the model, while IP-Adapter (Ye et al., 2023) incorporates image features as a condition through a decoupled cross-attention mechanism. Fine-tuning approaches have also been developed to incorporate new concepts into pretrained diffusion models by training on a minimal set of input images (Ruiz et al., 2023).

**Generating synthetic face recognition data.** Strict privacy regulations nowadays restrict the use and distribution of biometric data gathered without consent (Hoofnagle et al., 2019). Due to this, valuable datasets of web-scrapped face images are being retracted (Jasserand, 2022), which presents a challenge for developing face recognition models. As a solution, researchers are exploring creation of synthetic data with deep generative models (Boutros et al., 2023c). To enable control over various characteristics of generated faces, Deng et al. (2020) conditioned StyleGAN (Karras et al., 2019) on input 3D face priors. However, recognition models trained on the generated data achieved worse performance than those trained on real-world data. To tackle this, Qiu et al. (2021) introduced identity and domain mixup of synthetic and real data during training. Boutros et al. (2022) proposed to condition StyleGAN2 (Karras et al., 2020) on one-hot encoded identity labels. This improved intra-identity diversity at the cost of lowered inter-identity separability and a limited amount of possible identities. To address this, Tomašević et al. (2024) instead utilized identity features from a pretrained face recognition model as the condition, in addition to enabling the generation of multispectral data.

Recently, Boutros et al. (2023b) achieved the generation of identity-specific images with latent diffusion models by conditioning the denoising network on face recognition features. The proposed contextual partial dropout also prevented overfitting on identities and enabled control over inter-identity separability and intra-identity diversity. Differently, more recent approaches relied on pretrained diffusion models (Rombach et al., 2022) rather than training the models from scratch. Ruiz et al. (2023) presented the DreamBooth method that can associate a new identity to a rare text token through fine-tuning on images of the identity. During training, face images generated by the pretrained model are also used to preserve prior synthesis capabilities. Arc2Face (Papantoniou et al., 2024) instead replaces the identity token with recognition features and fine-tunes the model on a large-scale dataset. The textual-part of the prompt is also frozen so that control is tied primarily to the identity features, thus enabling more consistent generation of input identities. However, this comes at the cost of losing powerful prompt-based control. The recent IP-Adapter (Ye et al., 2023) has also been modified to use identity features as the condition while retaining control of text prompts through decoupled cross-attention. InstantID (Wang et al., 2024) extends these capabilities by incorporating spatial control with an auxiliary ControlNet-based (Zhang et al., 2023) module conditioned on facial landmarks and features. Despite advancements, identity consistency remained problematic, as the identity aspect was not considered in training objectives. To address this, Peng et al. (2024) introduced PortraitBooth, which incorporates an identity-based objective into the fine-tuning of DreamBooth (Ruiz et al., 2023). However, the solution only relies on the identity similarity of training images and generated noisy images, despite the success of more refined objectives on face recognition tasks (Trigueros et al., 2018). As a result, the approach tends to overfit on characteristics of training identities, e.g., the expression or pose, which lowers the diversity of produced images. Differently, our proposed ID-Booth fine-tuning framework utilizes a triplet objective that relies on the identity similarity between generated images and both training images (i.e., positive samples) and prior images produced by the initial model (i.e., negative samples). This enables better identity consistency while retaining the synthesis capabilities of pretrained latent diffusion models.

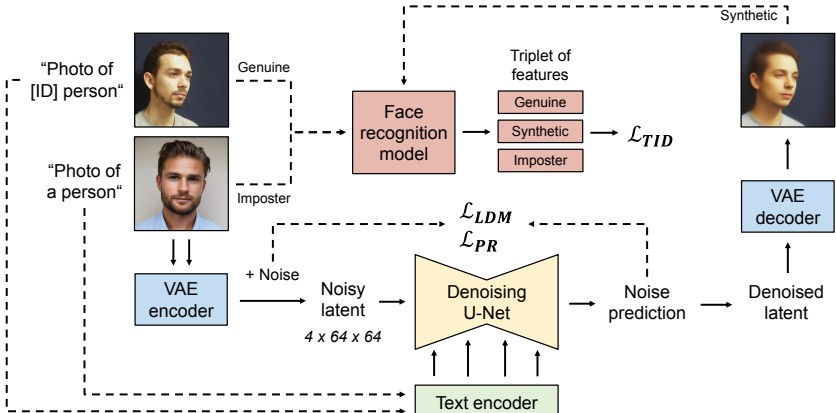

Figure 2: **Overview of the ID-Booth framework.** Fine-tuning of a pretrained diffusion model is performed with three training objectives. $\mathcal{L}_{LDM}$ and $\mathcal{L}_{PR}$ are aimed at the reconstruction of training and prior images. Differently, the proposed triplet identity objective $\mathcal{L}_{TID}$ focuses on the identity similarity between generated samples and both training and prior samples, to improve identity consistency without impacting the capabilities of the pretrained model.

## 3 METHODOLOGY

In the following sections, we delve into the inner-workings of latent diffusion models and existing approaches for fine-tuning them. Finally, we present the fine-tuning methodology of our proposed ID-Booth framework, which is showcased in Figure 2.

### 3.1 IMAGE GENERATION WITH LATENT DIFFUSION MODELS

Diffusion models (DMs) are a form of deep generative models that are trained to reverse a noising process that gradually degrades training images by adding noise at different scales. Denoising Diffusion Probabilistic Models (DDPMs) represent the most fundamental class of modern diffusion models (Sohl-Dickstein et al., 2015). They estimate the real data distribution from a noise-filled standard Gaussian distribution. This entails gradually denoising a noisy image $x_T \sim \mathcal{N}(0, \mathbf{I})$ to less noisy samples $x_t$ until a denoised data sample $x_0$ is reached.

First, we define the noising process in which a real data sample $x_0 \sim p(x_0)$ is corrupted into its noised versions $x_1, ..., x_T$ through a Markov chain of length $T$, as follows:

$$x_t = \mathcal{N}(\sqrt{\alpha_t}x_{t-1}, 1 - \alpha_t), \quad \forall t \in 1, ..., T, \tag{1}$$

where $\alpha_1, ..., \alpha_T$ represent a fixed variance schedule. Any step of the noised sample can also be efficiently produced directly from the input $x_0$ (Ho et al., 2020) as follows:

$$x_t = \sqrt{\bar{\alpha}_t}x_0, (1 - \bar{\alpha}_t)\epsilon, \tag{2}$$

with $\bar{\alpha}_t := \prod_{s=0}^{t} \alpha_s$, which enables uniform sampling of $t$ during training. The diffusion model learns to reverse the noising process with a denoising autoencoder $\epsilon_\theta(x_t, t)$, typically a U-Net network (Ronneberger et al., 2015), which predicts the noise $\epsilon$ that is added. The denoising network can then be trained by following the reweighted optimization objective (Ho et al., 2020):

$$\mathcal{L}_{DM} = \mathbb{E}_{x,\epsilon \sim \mathcal{N}(0,1),t}\big[\|\epsilon - \epsilon_\theta(x_t, t)\|_2^2\big]. \tag{3}$$

With recent Latent Diffusion Models (LDMs) (Rombach et al., 2022), the denoising process is instead carried out in the more efficient latent space of a pretrained Variational AutoEncoder (VAE) rather than the pixel space. The input sample $x_0$ is thus first mapped through the encoder model $\mathcal{E}$ to $z_0$, before the above-described operations are performed. In addition, the denoising process is also conditioned on encoded text prompts $c$ through the cross-attention mechanism, to improve control over the generation process. The training objective of LDMs can thus be defined as:

$$\mathcal{L}_{LDM} = \mathbb{E}_{z \sim \mathcal{E}(x), \epsilon \sim \mathcal{N}(0,1),t,c}\big[\|\epsilon - \epsilon_\theta(z_t, t, c)\|_2^2\big]. \tag{4}$$

Data generation can then be performed by randomly sampling a noisy sample $z_T$ in the latent space, denoising it with the predictor $\epsilon_\theta$ considering the provided prompt, and then mapping the denoised sample $z_0$ back to the pixel space through the VAE decoder $\mathcal{D}$.

### 3.2 Fine-tuning of diffusion models

Recent latent diffusion models provide unparalleled text-guided synthesis capabilities, owing to their training on various datasets of unprecedented scale (Rombach et al., 2022). However, their knowledge of very specific concepts and styles remains limited. This is also true for their ability to create images of a desired identity as prompting for a specific non-celebrity identity can be difficult or even impossible. To address this, Ruiz et al. (2023) propose to fine-tune a pretrained model on a small set of input images of a desired concept e.g. images of an identity, with the original training objective $\mathcal{L}_{LDM}$. However, this typically leads to overfitting on the input images and the loss of prior knowledge, e.g. the concept of what a person is. To prevent this, the authors introduce an additional training objective that is aimed at the preservation of prior knowledge. The pretrained diffusion model is first utilized to produce a set of prior images $x_{pr,0}$ related to the concept to be introduced, which are then used during training to retain the synthesis capabilities of the model. The proposed DreamBooth (Ruiz et al., 2023) approach, thus fine-tunes the model with $\mathcal{L}_{LDM}$ along with the following prior preservation objective:

$$\mathcal{L}_{PR} = \mathbb{E}_{z_{pr},c_{pr},\epsilon',t'}\left[\epsilon_{pr} - \epsilon_\theta(z_{pr,t'},t',c_{pr})\|_2^2\right], \tag{5}$$

where the $pr$ notation represents factors related to prior images generated with the initial model. In practice, fine-tuning of diffusion models is commonly performed by training solely the denoising network, while other components (e.g. the VAE and the text encoder) remain frozen. Methods often also rely on the use of the Low-rank adaptation method (LoRA) (Hu et al., 2022), which introduces new trainable layers at specific locations in the denoising network. During training only these layers are trained, leaving other pretrained weights unchanged. This facilitates faster training and more efficient storage of fine-tuned model weights, while still enabling the introduction of new concepts into the model or fine-tuning the model for a specific style.

### 3.3 Fine-tuning with identity-based objectives

Existing fine-tuning techniques have shown to be suitable for generating images of desired identities (Ruiz et al., 2023). However, the consistency of synthetic identities remains a prominent problem, both when considering consistency with desired input identities or synthetic identities in other generated samples. The likely cause are the training objectives, defined in Equations 4 and 5, which are focused solely on image reconstruction. To address this the identity aspect can also be incorporated into the training process through identity features extracted from face images with deep models for face recognition (Peng et al., 2024). However, the training of latent diffusion models does not entail the decoding of latent data back to the pixel space, since it is not required for either $\mathcal{L}_{LDM}$ or $\mathcal{L}_{PR}$. To produce suitable images at each step during training, the denoised latent $\hat{z}_0$ must first be estimated. This can be achieved using the predicted noise $\epsilon_\theta(z_t,t,c)$ and the noisy sample $z_t$ as follows:

$$\hat{z}_0 = \frac{z_t - \sqrt{1-\bar{\alpha}_t}\epsilon_\theta}{\sqrt{\bar{\alpha}_t}}. \tag{6}$$

Afterward, the estimated denoised latent $\hat{z}_0$ can be decoded to the estimated input image in the pixel space with $\hat{x}_0 = \mathcal{D}(\hat{z}_0)$. Then the facial region must be extracted with a face detector model for both the estimated and the input training image, denoted as $\hat{x}_0^f$ and $x_0^f$ respectively. Finally, the identity features for each image can be extracted with a face recognition model $\varphi$. A simple additional objective for training can then be constructed based on the cosine similarity $Sim$ of extracted identity features, as proposed with PortraitBooth (Peng et al., 2024):

$$\mathcal{L}_{ID} = 1 - Sim(\varphi(x_0^f), \varphi(\hat{x}_0^f)). \tag{7}$$

Despite its simplicity the objective is effective at guiding the diffusion model to better identity preservation. However, it can lead to overfitting on facial characteristics that might leak into the training identity embeddings, e.g. the expression or pose of subjects.

### 3.4 Triplet identity training objective

To address issues of previous methods, we propose to instead form a triplet objective based on the identity features extracted with a pretrained face recognition model $\varphi$. The proposed triplet identity objective $\mathcal{L}_{TID}$ utilizes identity features of the reconstructed sample $\hat{x}_0$ as the anchor, the input

image $x_0$ as a positive example of an identity and the prior images $x_{pr,0}$ as a negative example. Formally, our triplet identity objective can be defined as follows:

$$\mathcal{L}_{TID} = max\{Sim(\varphi(x_0^f), \varphi(\hat{x}_0^f)) - Sim(\varphi(x_{pr,0}^f), \varphi(\hat{x}_0^f)) + m, 0\}, \tag{8}$$

where the notations introduced before apply. In addition, $m$ represents a non-negative margin, i.e. the minimum difference between positive and negative similarities that is required for the loss to be zero. Compared to $\mathcal{L}_{ID}$, employing a triplet-based objective reduces the risk of overfitting on unintentional characteristics of training identities as they are also present in negative examples. Overall, our proposed ID-Booth framework utilizes the following training objective for fine-tuning:

$$\mathcal{L}_{Total} = \mathcal{L}_{LDM} + \mathcal{L}_{PR} + \mathcal{L}_{TID}, \tag{9}$$

as illustrated in Figure 2. The framework is designed to improve identity consistency through fine-tuning while retaining the synthesis capabilities of pretrained models.

## 4 EXPERIMENTS AND RESULTS

**Datasets.** To fine-tune the diffusion models we utilize the Tufts Face Database (TFD) (Panetta et al., 2018), which contains images captured in a constrained laboratory setting with the consent of subjects. Following the preprocessing steps outlined in the supplementary material, the dataset comprises 2213 images of 105 subjects. To evaluate the generated images we also rely on the Flickr Faces High-Quality (FFHQ) (Karras et al., 2019) dataset of $70,000$ diverse in-the-wild face images.

**Implementation.** We evaluate the suitability of our fine-tuning method on two pretrained diffusion models, Stable Diffusion 2.1 (SD-2.1) (Rombach et al., 2022) and its successor Stable Diffusion XL (SD-XL) (Podell et al., 2024), with the LoRA method (Hu et al., 2022). For fine-tuning we utilize the objectives specified by either DreamBooth (Ruiz et al., 2023) (i.e. $\mathcal{L}_{LDM} + \mathcal{L}_{PR}$), PortraitBooth (Ruiz et al., 2023) (i.e. $\mathcal{L}_{LDM} + \mathcal{L}_{PR} + \mathcal{L}_{ID}$) or our proposed ID-Booth objective i.e. $\mathcal{L}_{Total}$. Here, the identity objectives are based on identity features extracted with the pretrained ArcFace recognition model (Deng et al., 2019) from face regions of noisy samples detected with the the Multi-Task Cascaded Convolutional Neural Network (MTCNN) (Zhang et al., 2016). Fine-tuning is performed with images of each identity from TFD (Panetta et al., 2018) along with 200 prior preservation face images generated with the pretrained models. With each model we generate 21 images per identity through 30 denoising steps and a guidance scale of 5.0, either of a resolution $512 \times 512$ with SD-2.1 or $1024 \times 1024$ with SD-XL. This is done either with a prompt that defines a close-up portrait image of an identity (denoted as *Base*) or a prompt that in addition specifies the expression of the subject and the environment surrounding the subject (denoted as *Complex*). Additional implementation details and the utilized prompts are available in the supplementary material.

**Evaluation methodology.** We evaluate our proposed framework based on images generated by the fine-tuned models. For a fair comparison with TFD and FFHQ, the produced images, whose facial regions are often smaller, are first aligned and cropped to $112 \times 112$ based on face landmarks detected by MTCNN (Zhang et al., 2016). The quality of images is then determined with Fréchet Inception Distance (FID) (Heusel et al., 2017) and CLIP Maximum Mean Discrepancy (CMMD) (Jayasumana et al., 2024), while improved precision and accuracy are used to measure the fidelity and diversity of images, respectively (Kynkäänniemi et al., 2019). These measures operate by comparing distributions of synthetic and real-world data via image features of pretrained vision models (e.g., Inception-v3 (Szegedy et al., 2016)). Differently, Certainty Ratio Face Image Quality Assessment (CR-FIQA) (Boutros et al., 2023a) evaluates the relative classifiability and in turn quality of each face image individually with a pretrained ResNet-101 backbone (He et al., 2016). Furthermore, we investigate intra-identity consistency and inter-identity separability with genuine and imposter distributions, formed by pairs of identity features from the ArcFace model (Deng et al., 2019). To this end, we report the mean and standard deviation of distributions along with established metrics, including Equal Error Rate (EER), False Match Rate at a false non-match rate of $1.0\%$ (FMR100) or $0.01\%$ (FMR1000) and the Fisher Discriminant Ratio (FDR) (ISO/IEC 19795-1:2021). Lastly, we use the produced images to train a small-scale ResNet-18 CosFace recognition model (Wang et al., 2018) and evaluate its performance on five state-of-the-art verification benchmarks, including Labeled Faces in the Wild (LFW) (Huang et al., 2007), its Cross-Age and Cross-Pose subsets CA-LFW (Zheng et al., 2017) and CP-LFW (Zheng & Deng, 2018), Celebrities in Frontal-Profile in the Wild (CFP-FP) (Sengupta et al., 2016) and AgeDB-30 (Moschoglou et al., 2017).

Table 1: **Evaluation of quality, fidelity, and diversity of image samples.** Image quality is assessed with FID (Heusel et al., 2017) and CMMD (Jayasumana et al., 2024) scores, while fidelity and diversity are measured with precision and recall, respectively (Kynkäänniemi et al., 2019). These measures compare synthetic distributions to real-world data of either TFD (Panetta et al., 2018) or FFHQ (Karras et al., 2019). Meanwhile, CR-FIQA (Boutros et al., 2023a) measures the face image quality of each synthetic sample separately so the mean and standard deviation are reported.

| Data from | Prompt | Fine-tuning | FID ↓ | | CMMD ↓ | | Precision ↑ | | Recall ↑ | | CR-FIQA ↑ |
| | | | TFD | FFHQ | TFD | FFHQ | TFD | FFHQ | TFD | FFHQ | − |
|---|---|---|---|---|---|---|---|---|---|---|---|
| TFD | – | – | 17.446 | 79.884 | 0.008 | 0.929 | 0.507 | 0.684 | 0.316 | 0.003 | 2.131 ± 0.093 |
| FFHQ | – | – | 79.884 | 10.425 | 0.929 | 0.005 | 0.684 | 0.786 | 0.003 | 0.781 | 2.089 ± 0.145 |
| SD-2.1 | Base | – | 98.010 | 79.240 | 1.485 | 0.923 | 0.002 | 0.413 | 0.356 | 0.317 | 1.737 ± 0.643 |
| | | DreamBooth | 35.285 | 61.511 | 0.530 | **1.242** | 0.178 | **0.614** | 0.462 | 0.040 | 2.079 ± 0.351 |
| | | PortraitBooth | **32.991** | **60.551** | 0.519 | 1.260 | **0.191** | 0.581 | 0.469 | **0.049** | **2.109 ± 0.298** |
| | | ID-Booth | 33.488 | 61.000 | 0.519 | 1.255 | 0.191 | 0.570 | **0.477** | 0.038 | 2.097 ± 0.321 |
| | Complex | – | 89.890 | 44.860 | 1.514 | 1.048 | 0.000 | 0.513 | 0.497 | 0.475 | 1.857 ± 0.521 |
| | | DreamBooth | 65.758 | 51.901 | 1.432 | 1.115 | 0.001 | 0.597 | **0.438** | **0.186** | 1.991 ± 0.487 |
| | | PortraitBooth | **62.038** | 51.912 | 1.412 | 1.104 | **0.004** | 0.614 | 0.332 | 0.147 | **2.028 ± 0.452** |
| | | ID-Booth | 62.114 | **51.815** | **1.407** | **1.103** | 0.001 | **0.622** | 0.320 | 0.184 | 2.019 ± 0.471 |
| SD-XL | Base | – | 91.633 | 74.058 | 3.236 | 2.379 | 0.000 | 0.464 | 0.106 | 0.225 | 1.986 ± 0.381 |
| | | DreamBooth | **40.806** | **69.788** | 0.812 | 1.482 | **0.155** | **0.543** | **0.306** | 0.007 | **2.178 ± 0.106** |
| | | PortraitBooth | 41.664 | 72.679 | **0.807** | **1.454** | 0.135 | 0.515 | 0.226 | 0.007 | 2.175 ± 0.102 |
| | | ID-Booth | 41.637 | 72.865 | 0.809 | 1.458 | 0.129 | 0.531 | 0.175 | **0.009** | 2.175 ± 0.097 |
| | Complex | – | 86.399 | 41.411 | 2.328 | 1.590 | 0.001 | 0.663 | 0.314 | 0.296 | 2.083 ± 0.210 |
| | | DreamBooth | 67.919 | **51.590** | 0.722 | **0.982** | 0.015 | **0.494** | 0.488 | **0.155** | 2.139 ± 0.228 |
| | | PortraitBooth | 66.278 | 53.502 | 0.710 | 0.999 | 0.016 | 0.466 | **0.531** | 0.102 | 2.141 ± 0.227 |
| | | ID-Booth | 66.259 | 53.488 | **0.709** | 0.999 | **0.017** | 0.459 | 0.493 | 0.137 | **2.144 ± 0.220** |

(↓) – Lower is better; (↑) – Higher is better; (**Bold**) – Best fine-tuning result; (Underline) – Second best fine-tuning result

## 4.1 Evaluation of generated images

**Image quality.** We begin our evaluation by assessing the overall quality of images, produced by either DreamBooth (Ruiz et al., 2023), PortraitBooth (Peng et al., 2024) or the proposed ID-Booth framework, in terms of FID (Heusel et al., 2017), CMMD (Jayasumana et al., 2024) and CR-FIQA (Boutros et al., 2023a). From results reported in Table 1, we can observe that the data generated with base prompts better resembles the real-world constrained-setting images of TFD (Panetta et al., 2018). Meanwhile data generated with complex prompts, which define the environment and expression, better matches the in-the-wild images of FFHQ (Karras et al., 2019). Results also reveal that all fine-tuning approaches increase the image quality of initial pretrained models, likely due to the increased image diversity of initial models, which often generate images that do not contain the entire face. In addition, we note that SD-2.1 has problems with unnatural face artifacts especially with complex prompts, as exhibited by lower CR-FIQA scores with a drastically higher standard deviation than SD-XL. Importantly, we observe notable differences between models trained with or without identity-based objectives. This is supported by CMMD and CR-FIQA scores, where PortraitBooth and ID-Booth both achieve better quality than DreamBooth. In comparison, the difference between PortraitBooth or ID-Booth tends to be minimal. Overall, our proposed ID-Booth framework does not negatively impact the quality of generated images and often even achieves better quality than existing fine-tuning approaches. Figures 1 and 4 allow for a qualitative evaluation of samples generated by a fine-tuned SD-XL model with complex prompts.

**Fidelity and diversity.** Next, we analyse the produced images in terms of fidelity, i.e., the degree to which they resemble real samples, and diversity, i.e., how well they cover the variability of real samples (Sajjadi et al., 2018). To this end, we rely on the precision and the accuracy metric, respectively (Kynkäänniemi et al., 2019), in addition to qualitative samples in Figures 1 and 4. With both diffusion models base prompts achieve drastically better precision on TFD (Panetta et al., 2018), as they tend to generate subjects with a neutral expression in a constrained setting similar to the real-world images. Meanwhile, complex prompts result in better recall on FFHQ (Karras et al., 2019), as they facilitate the generation of more diverse images. SD-2.1 often attains better precision and recall than SD-XL, however, this likely due to less consistent quality and possible artifacts, as reported by quality-based measures. Interestingly, fine-tuning the pretrained models often results in better precision and recall not only on the training TFD images but in certain cases even on FFHQ. Importantly, when comparing the different fine-tuning methods, we observe that PortraitBooth (Peng et al., 2024) achieves drastically worse recall with complex prompts on the FFHQ dataset than Dream-

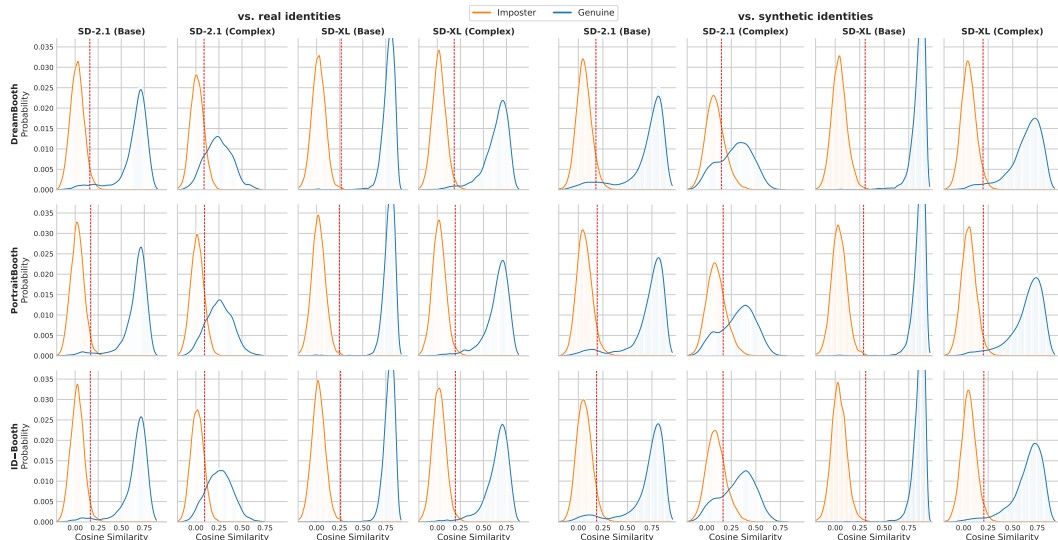

Figure 3: **Plots of genuine and imposter distributions either between synthetic and real-world samples or only among synthetic samples.** Distributions are based on the cosine similarity between identity features of synthetic samples and either samples from the corresponding identity (genuine pair) or a different identity (imposter pair), from either the real-world TFD dataset (Panetta et al., 2018) or the same synthetic dataset. For each dataset, all possible genuine pairs are formed, along with an equal amount of randomly sampled imposter pairs. Identity features are obtained with the pretrained ArcFace recognition model (Deng et al., 2019).

Booth (Ruiz et al., 2023). This drop in diversity may be attributed to lower prompt adherence, as seen in Figure 1, where compared to DreamBooth, PortraitBooth subjects tend to lose the desired expression or default to a front-facing pose. In comparison, our proposed ID-Booth framework does not display the same issues with complex prompts, as it achieves notably higher recall scores, more similar to DreamBooth (Ruiz et al., 2023), as also observed in Figure 1. This ability for generating diverse images is crucial for creating synthetic datasets suitable for training face recognition models.

### 4.2 RECOGNITION-BASED EXPERIMENTS

**Identity consistency and separability.** To determine the suitability of generated images for forming face recognition datasets we must also examine the consistency and separability of identities in the images. To this end, we first analyse the similarity of synthetic identities to either their corresponding or a different training identity, based on identity features from the ArcFace recognition model Deng et al. (2019). From genuine and imposter distributions on the left in Figure 3 and verification results in Table 2, we can observe that the SD-2.1 model achieves notably worse identity consistency, i.e. the similarity between corresponding synthetic and real identities, across all scenarios than the SD-XL model. The same can be observed for inter-identity separability, i.e. the similarity between different synthetic and real identities, as the overlap between imposter and genuine distributions is larger. This is especially true when utilizing complex prompts, which highly affect the identities generated with SD-2.1. Importantly, results indicate that employing identity-based objectives greatly improves both consistency and separability between synthetic and real identities. Fine-tuning with the proposed ID-Booth framework ensures comparable results to PortraitBooth (Peng et al., 2024) when paired with SD-2.1, while providing notable improvements with SD-XL. Figure 4 further demonstrates that ID-Booth achieves better identity consistency than DreamBooth (Ruiz et al., 2023), while maintaining better text-based control over the generation process and in turn higher image diversity than PortraitBooth (Peng et al., 2024).

Furthermore, we investigate the similarity among synthetic samples of the same identity and the similarity among samples of different synthetic identities. Distributions on the right in Figure 3 as well as results in Table 2 reveal a similar trend as before. Notably, ID-Booth achieves the highest consistency among generated samples of the same identity and the largest separability between synthetic

Table 2: **Evaluation of identity consistency and separability between synthetic and real-world identities.** Reported are verification measures of genuine and imposter distributions in Figure 3.

| | Data from | Prompt | Fine-tuning | EER↓ | FMR100↓ | FMR1000↓ | Imposter $\mu \pm \sigma$↓ | Genuine $\mu \pm \sigma$↑ | FDR↑ |
|---|---|---|---|---|---|---|---|---|---|
| | TFD | – | – | 0.001 | 0.001 | 0.001 | 0.021 ± 0.073 | 0.871 ± 0.065 | 75.753 |
| vs. real identities | SD-2.1 | Base | DreamBooth | 0.039 | 0.052 | 0.066 | **0.022 ± 0.075** | 0.638 ± 0.170 | 10.916 |
| | | | PortraitBooth | **0.029** | **0.034** | **0.041** | 0.024 ± 0.073 | **0.653 ± 0.148** | **14.531** |
| | | | ID-Booth | 0.031 | 0.038 | 0.044 | 0.023 ± 0.073 | 0.650 ± 0.154 | 13.541 |
| | | Complex | DreamBooth | 0.153 | 0.364 | 0.500 | **0.014 ± 0.072** | 0.244 ± 0.144 | 2.030 |
| | | | PortraitBooth | 0.137 | **0.322** | 0.489 | 0.015 ± 0.072 | 0.255 ± 0.142 | **2.268** |
| | | | ID-Booth | **0.137** | 0.326 | **0.465** | 0.016 ± 0.071 | 0.254 ± 0.141 | 2.260 |
| | SD-XL | Base | DreamBooth | 0.002 | 0.002 | 0.002 | 0.022 ± 0.074 | 0.782 ± 0.071 | 54.952 |
| | | | PortraitBooth | 0.003 | 0.003 | 0.003 | 0.021 ± 0.075 | 0.786 ± 0.074 | 53.233 |
| | | | ID-Booth | **0.002** | **0.002** | **0.002** | **0.021 ± 0.074** | **0.786 ± 0.067** | **58.578** |
| | | Complex | DreamBooth | 0.019 | 0.023 | 0.035 | 0.019 ± 0.074 | 0.635 ± 0.144 | 14.537 |
| | | | PortraitBooth | 0.016 | 0.018 | 0.031 | 0.019 ± 0.074 | 0.646 ± 0.138 | 16.087 |
| | | | ID-Booth | **0.015** | **0.016** | **0.024** | 0.019 ± 0.074 | **0.647 ± 0.135** | **16.473** |
| vs. synthetic identities | SD-2.1 | Base | DreamBooth | 0.067 | 0.090 | 0.106 | **0.057 ± 0.079** | 0.684 ± 0.224 | 6.955 |
| | | | PortraitBooth | 0.052 | 0.063 | 0.071 | 0.062 ± 0.077 | **0.713 ± 0.196** | **9.596** |
| | | | ID-Booth | 0.061 | 0.073 | 0.082 | 0.061 ± 0.079 | 0.702 ± 0.209 | 8.224 |
| | | Complex | DreamBooth | 0.242 | 0.596 | 0.803 | **0.087 ± 0.098** | 0.285 ± 0.174 | 0.985 |
| | | | PortraitBooth | 0.227 | 0.544 | 0.766 | 0.097 ± 0.100 | **0.314 ± 0.176** | 1.138 |
| | | | ID-Booth | **0.226** | **0.529** | **0.734** | 0.096 ± 0.099 | 0.312 ± 0.177 | **1.139** |
| | SD-XL | Base | DreamBooth | 0.002 | 0.002 | 0.002 | 0.040 ± 0.075 | 0.851 ± 0.078 | 56.614 |
| | | | PortraitBooth | 0.003 | 0.003 | 0.003 | 0.037 ± 0.075 | 0.856 ± 0.073 | 61.196 |
| | | | ID-Booth | **0.001** | **0.001** | **0.002** | **0.037 ± 0.075** | **0.856 ± 0.066** | **67.493** |
| | | Complex | DreamBooth | 0.037 | 0.052 | 0.078 | 0.051 ± 0.077 | 0.629 ± 0.176 | 9.045 |
| | | | PortraitBooth | 0.035 | 0.047 | 0.072 | 0.050 ± 0.078 | 0.643 ± 0.173 | 9.715 |
| | | | ID-Booth | **0.031** | **0.040** | **0.063** | 0.050 ± 0.078 | **0.648 ± 0.167** | **10.492** |

(↓) – Lower is better; (↑) – Higher is better; (**Bold**) – Best result; (Underline) – Second best result

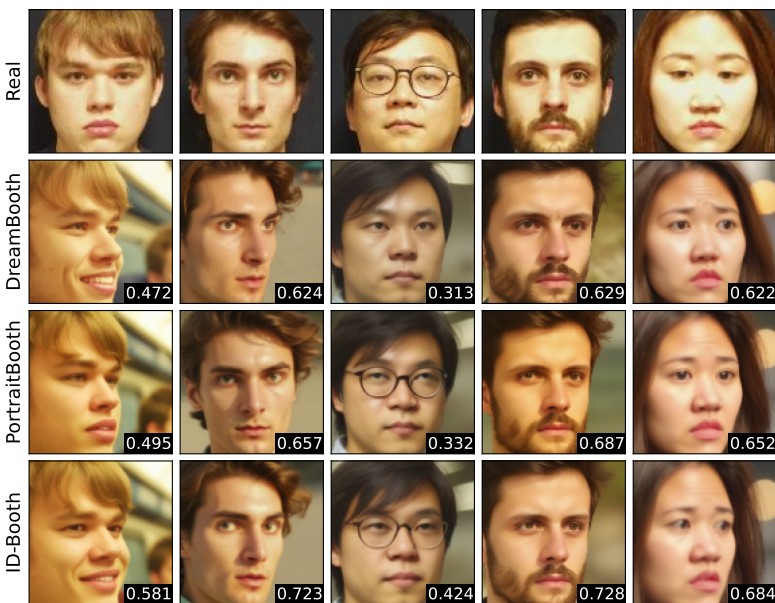

Figure 4: **Comparison of training and generated identities.** ID-Booth achieves better identity consistency than DreamBooth (Ruiz et al., 2023) while retaining better prompt adherence and diversity of the pretrained SD-XL (Podell et al., 2024) than PortraitBooth (Peng et al., 2024). Reported is the cosine similarity of identity features extracted with the ArcFace model (Deng et al., 2019).

samples of different identities. Overall, the presented results showcase that the proposed ID-Booth fine-tuning framework drastically improves the ability to generate consistent desired identities with pretrained diffusion models. This aspect is important for ensuring the generation of privacy preserving synthetic datasets, which contain only identities that match those in the training set, for which we have consent from subjects.

Table 3: **Verification performance of recognition models trained on different synthetic datasets.** Reported is the accuracy of a trained CosFace model (Wang et al., 2018) on 5 real-world verification benchmarks. During training the LFW benchmark (Huang et al., 2007) is used for validation.

| Training setting | | | Val. ↑ | Verification accuracy on benchmarks ↑ | | | | |
|---|---|---|---|---|---|---|---|---|
| Data from | Prompt | Fine-tuning | LFW | AgeDB-30 | CA-LFW | CFP-FP | CP-LFW | Average |
| TFD | – | – | 0.672 | 0.501 | 0.548 | 0.598 | 0.542 | $0.547 \pm 0.034$ |
| SD-2.1 | Base | DreamBooth | **0.665** | **0.525** | 0.539 | 0.572 | 0.542 | $0.544 \pm 0.017$ |
| | | PortraitBooth | 0.664 | 0.507 | 0.532 | **0.602** | 0.548 | $0.547 \pm 0.035$ |
| | | ID-Booth | 0.664 | 0.509 | **0.541** | 0.579 | **0.565** | **$0.548 \pm 0.027$** |
| | Complex | DreamBooth | 0.681 | 0.499 | **0.553** | 0.591 | **0.565** | $0.552 \pm 0.034$ |
| | | PortraitBooth | **0.682** | 0.492 | 0.551 | **0.615** | 0.552 | **$0.553 \pm 0.043$** |
| | | ID-Booth | 0.668 | **0.500** | 0.537 | 0.602 | 0.561 | $0.550 \pm 0.037$ |
| SD-XL | Base | DreamBooth | 0.674 | 0.515 | 0.550 | 0.582 | 0.540 | $0.547 \pm 0.024$ |
| | | PortraitBooth | 0.679 | 0.491 | **0.558** | 0.611 | **0.555** | $0.554 \pm 0.042$ |
| | | ID-Booth | **0.688** | **0.529** | 0.550 | **0.611** | 0.539 | **$0.557 \pm 0.032$** |
| | Complex | DreamBooth | **0.745** | 0.496 | 0.579 | **0.615** | **0.579** | $0.567 \pm 0.044$ |
| | | PortraitBooth | 0.726 | 0.507 | 0.584 | 0.608 | 0.569 | $0.567 \pm 0.037$ |
| | | ID-Booth | 0.732 | **0.532** | **0.599** | 0.605 | 0.567 | **$0.575 \pm 0.029$** |

(↑) – Higher is better; (**Bold**) – Best result; (Underline) – Second best result

**Training face recognition models.** Finally, we also explore the utility of the generated data in a real-world scenario, namely for training deep face recognition models. To this end, we train a Cos-Face recognition model (Wang et al., 2018) on the synthetic datasets and evaluate its performance on state-of-the-art face verification benchmarks. From results reported in Table 3 we can discern that training data generated by SD-XL enables better verification performance than data of the SD-2.1 model. A notable improvement can also be observed when training on data generated with complex prompts, due to the higher diversity of images. Importantly, our method produces training data, which results in recognition models that achieve the highest average verification accuracy across all benchmarks. This is especially evident with SD-XL on the AgeDB-30 benchmark (Moschoglou et al., 2017), likely due to the improved diversity of images and identity consistency that our method provides compared to existing approaches. Furthermore, with our proposed method, we achieve drastically better verification performance than when training recognition models on the real-world Tufts Face Database (TFD) (Panetta et al., 2018), despite the similar scale in terms of the number of identities and images in our produced synthetic datasets.

## 5 CONCLUSION

In this paper, we presented ID-Booth, a new framework for fine-tuning pretrained diffusion models to facilitate the generation of diverse high-quality identity-consistent images. To this end, ID-Booth relies on a novel triplet identity training objective that improves both intra-identity consistency and inter-identity separability, while better retaining the image diversity of pretrained models. This applies when exploring identity similarity either between synthetic and real images or only among synthetic ones. Throughout the experiments, we demonstrate the suitability of our fine-tuning framework on two state-of-the-art diffusion models with text prompts of varying complexity. In addition, we showcase that training recognition models on data produced by our method results in better performance across five verification benchmarks than when utilizing a real-world dataset of similar scale or synthetic datasets of existing approaches. Overall, the ID-Booth framework presents a potential solution for creating diverse privacy-preserving recognition datasets based on existing small-scale training datasets collected with suitable consent. However, our work also highlights the challenges with training recognition models on privacy-preserving datasets. With regards to future work, we aim to investigate the applicability of identity-based objectives in the training of conditioning approaches and exploring the creation of larger-scale datasets.

## 6 ACKNOWLEDGEMENTS

The acknowledgements have been omitted for the purposes of the double-blind review process.

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
