SUPPLEMENTARY MATERIAL

## A    REAL-WORLD DATASETS

For the purposes of fine-tuning the diffusion models we utilize images of the Tufts Face Database (TFD) (Panetta et al., 2018). In total, the dataset contains over $10,000$ images of 113 human subjects captured in a constrained setting across various light spectra. We focus on images captured with four visible field cameras under constant diffused light in a semi-circle around the subjects. During preprocessing, we remove heavily-blurred images and side-profile images, which lack key facial features (e.g. two eyes) and then crop the images to focus on the face region. These steps result in a dataset of 2113 images of 105 subjects. Next we use the eye landmarks, detected with the Multi-Task Cascaded Convolutional Neural Network (MTCNN) (Zhang et al., 2016) to define an affine transform, which improves the alignment and size consistency of faces across the dataset. Finally, we resize the images to either a resolution of $512 \times 512$ or $1024 \times 1024$, depending on the image generation model to be trained. Throughout the experiments, TFD is also used for evaluating the synthesis capabilities of the models, along with the Flickr Faces HQ (FFHQ) (Karras et al., 2019) dataset, which contains $70,000$ high-quality in-the-wild face images with varied ethnicity, expressions, lighting and environments. Characteristics of both datasets are summarized in Table 4.

Table 4: **Overview of utilized face image datasets.** TFD (Panetta et al., 2018) is used for fine-tuning and later validation along with FFHQ (Karras et al., 2019). Other datasets form verification benchmarks for evaluating recognition models trained on the generated data.

| Dataset | #Images | #IDs | Resolution | Purpose |
|---|---|---|---|---|
| TFD (Panetta et al., 2018) | $> 10,000$ | 113 | $3280 \times 2464$ | – |
| TFD* (Panetta et al., 2018) | 2213 | 105 | $256 \times 256$ | FT & SV |
| FFHQ (Karras et al., 2019) | $70,000$ | N/A | $1024 \times 1024$ | SV |
| LFW (Huang et al., 2007) | $13,233$ | 5749 | $250 \times 250$ | REC |
| CA-LFW (Zheng et al., 2017) | 7156 | 2996 | $250 \times 250$ | REC |
| CP-LFW (Zheng & Deng, 2018) | 5984 | 2296 | $250 \times 250$ | REC |
| AgeDB-30 (Moschoglou et al., 2017) | $16,488$ | 568 | Various | REC |
| CFP-FP (Sengupta et al., 2016) | 7000 | 500 | Various | REC |

(*) – Preprocessed dataset; (FT) – Fine-tuning; (SV) – Synthesis validation
(REC) – Verification experiments;

## B    IMPLEMENTATION DETAILS

**Diffusion models and fine-tuning**    In our work, we rely on latent diffusion models for image generation, specifically on two models from the open-source state-of-the-art framework known as Stable Diffusion. This includes Stable Diffusion 2.1 (SD-2.1) (Rombach et al., 2022) and its successor Stable Diffusion XL (SD-XL) (Podell et al., 2024), which are capable of generating high-quality and diverse images at a resolution of $512 \times 512$ and $1024 \times 1024$ respectively. For both models we utilize the discrete denoising scheduler with $\beta_{end} = 0.012$, $beta_{start} = 8.5 \times 10^{-4}$ and 1000 denoising timesteps (Ho et al., 2020).

Throughout the experiments we fine-tune the two models on images of each identity in the Tufts Face Database (Panetta et al., 2018). To this end, we utilize the training objective either defined by DreamBooth (Ruiz et al., 2023), i.e., $\mathcal{L}_{LDM} + \mathcal{L}_{PR}$, PortraitBooth (Ruiz et al., 2023), i.e., $\mathcal{L}_{LDM} + \mathcal{L}_{PR} + \mathcal{L}_{ID}$, or by our proposed ID-Booth framework, i.e., $\mathcal{L}_{Total} = \mathcal{L}_{LDM} + \mathcal{L}_{PR} + \mathcal{L}_{TID}$. However, differently from previous previous methods (Peng et al., 2024), we rely on the detection of faces as the decision factor for which images are suitable for the identity-based training objectives $\mathcal{L}_{ID}$ and $\mathcal{L}_{TID}$, rather than utilizing a hard-coded threshold at a specific denoising step. In addition, we rely on the Low-Rank Adaptation (LoRA) (Hu et al., 2022) method during fine-tuning, which freezes the initial diffusion model but introduces new layers to it that are trained instead, thus minimizing the effect on the synthesis capabilities of the model. Specifically, we add two linear layers with a rank of $4$ to each of the cross-attention blocks, which are initialised with a Gaussian distribution.

Before training, we also generate 200 images with the initial pretrained diffusion models and the prompt `photo of a person`. These prior images are used for preservation of prior concepts

through $\mathcal{L}_{PR}$ during training, following the DreamBooth method (Ruiz et al., 2023). We then perform fine-tuning with images of a desired identity and the prompt `photo of [ID] person`, where `[ID]` represents the token that will be tied to the new identity. For this purpose we rely on a rare text token, namely `sks`, following existing works (Ruiz et al., 2023). For fine-tuning we utilize an initial learning rate of $10^{-4}$ and the AdamW optimizer (Loshchilov & Hutter, 2017) with $\beta_1 = 0.9$, $\beta_2 = 0.999$, $\epsilon = 10^{-8}$ and a weight decay of 0.01. In addition, we rely on the half-precision floating point format to lower VRAM usage. The fine-tuning process is stopped either after 4000 steps (i.e. 20 epochs) with SD-2.1 or 1000 steps (i.e. 5 epochs) with SD-XL, based on our initial observations when experimenting with the models and existing works (Ruiz et al., 2023; Peng et al., 2024).

**Image generation**    Each fine-tuned SD model is then used to generate 21 images per identity, based on the average amount of images in the Tufts Face Database (Panetta et al., 2018). Data generation is performed with a guidance scale of 5.0 and 30 inference steps with the same discrete denoising scheduler as during training. The goal is to generate diverse synthetic images of desired identities under various scenarios. To this end, we define two variants of text prompts for conditioning the diffusion models. The first prompt (denoted as *Base* in the experiments) is used as a baseline to produce portrait images of a desired identity that are similar to those in the training set:

```
photo of [ID] person, close-up portrait
```

To produce images that resemble real-world in-the-wild datasets (Karras et al., 2019), we utilize a second variant of prompts (denoted as *Complex*), which also define the environment that the image is taken in and the expression of the subject:

```
photo of [ID] person, close-up portrait, busy [B] environment,
    [E] expression
```

Here `[B]` and `[E]` represent possible environments and expressions randomly selected from predefined lists. To produce diverse images in terms of lighting conditions, clothes and overall style we define the following list of environments:

```
[office, bus, forest, laboratory, factory, beach, construction
    site, hospital, city street, night club]
```

Meanwhile the expression `[E]` is selected from a shorter list of basic human emotions used throughout existing works on emotion recognition (Canal et al., 2022):

```
[neutral, happy, sad, angry, shocked]
```

In addition, we rely on a negative prompt to ensure the generation of more realistic images suitable for training face recognition models that are used in real-world scenarios:

```
cartoon, cgi, render, illustration, painting, drawing, black and
    white, bad body proportions, landscape
```

To address issues with the SD-2.1 model (Ho et al., 2020) not producing the correct gender, we also add the descriptor `female` or `male` before the token `person` to the SD-2.1 prompts.

**Recognition experiment details.**    During the experiments we also explore the suitability of the produced synthetic data for training face recognition models. To this end, we train a small-scale ResNet-18 recognition model (Wang et al., 2018) on the different synthetic datasets. For training we utilize the CosFace loss function (Wang et al., 2018) and the Stochastic Gradient Descent (SGD) optimizer with 0.9 momentum and a weight decay of $5 \times 10^{-4}$. The learning rate is initially set to 0.1 and is lowered by a factor of 10 after the 22nd, the 30th, and the 35th epoch. During training 4 random augmentations with a magnitude of 16 are applied to training images and a dropout ratio of 0.4 is used. Training is stopped once no improvement across 5 epochs is observed on the LFW (Huang et al., 2007) benchmark.

**Experimental hardware**    The experiments were conducted on a Desktop PC with an AMD Ryzen 7 7800X3D CPU with 128 GB of RAM and an Nvidia RTX 4090 GPU with 24 GB of video RAM.

## C    EVALUATION METHODOLOGY

**Measuring image quality, fidelity and diversity.**    To evaluate the synthesis capabilities of the
fine-tuned models we compare the produced images with both the Tufts Face Database (Panetta
et al., 2018) and the more diverse FFHQ dataset (Karras et al., 2019). To this end, we utilize the
following performance measures:

- Fréchet Inception Distance (FID) (Heusel et al., 2017), which estimates the overall quality
  of synthetic images. This is achieved by estimating the difference between distributions of
  image features extracted from the real and the synthetic dataset with a pretrained Inception-
  v3 model (Szegedy et al., 2016). Lower scores imply better correspondence.
- CLIP Maximum Mean Discrepancy (CMMD) (Jayasumana et al., 2024) presents an al-
  ternative quality measure to FID (Heusel et al., 2017) as it offers a different perspec-
  tive through features extracted with the Contrastive Language Image Pre-training (CLIP)
  model (Radford et al., 2021). By measuring the squared maximum mean discrepancy it
  also addresses inconsistencies of FID (Heusel et al., 2017) on small datasets.
- Certainty Ratio Face Image Quality Assessment (CR-FIQA) measure (Boutros et al.,
  2023a), which is designed specifically for evaluating the quality of face images. It mea-
  sures the quality through the relative classifiability of a given face image with a pretrained
  ResNet-101 network (He et al., 2016).
- Precision and Recall (Kynkäänniemi et al., 2019), which measure the fidelity and diver-
  sity, respectively, by considering the distance between nearest neighbour embeddings of
  images extracted with the Inception-v3 network (Szegedy et al., 2016) pretrained on Ima-
  geNet (Deng et al., 2009).

Here, it should be noted that the fine-tuned diffusion models produce images that often contain more
context than just the face region, differently from the FFHQ dataset (Karras et al., 2019). Thus,
to a allow for a fair evaluation of specifically the face region we preprocess the generated images,
following the preprocessing steps of FFHQ (Karras et al., 2019). This includes first detecting facial
landmarks with the Multi-Task Cascaded Convolutional Neural Network (MTCNN) (Zhang et al.,
2016) and then defining an affine transform to align them to a set of predefined positions. Finally,
images are cropped to a resolution of $112 \times 112$, suitable for the CosFace recognition model (Wang
et al., 2018). For FID, CMMD as well as precision and recall, which utilize both synthetic and real-
world distributions for evaluation, we utilize the entire synthetic datasets and the entire TFD dataset,
but 2500 randomly sampled images from FFHQ. Furthermore, when comparing images within the
real-world datasets, we randomly split the TFD dataset in half to form two distributions and also
randomly sample 5000 images from FFHQ, which are then split into two distributions of 2500.

**Assesment of identity consistency and separability.**    We also investigate the generated images
in terms of the identity aspect in order to better understand the consistency and separability of
identities of generated datasets. For this purpose we utilize genuine and imposter score distribution
plots, based on the cosine similarity of features extracted with the pretrained ArcFace recognition
model (Deng et al., 2019). Alongside we provide results from the following verification measures:

- Equal Error Rate (EER) (Maio et al., 2002), which is the point on the Receiver Operating
  Characteristics (ROC) curve, where the False Match Rate (FMR) equals the False Non-
  Match Rate (FNMR).
- FMR100 and FMR1000, which report the lowest the False Non-Match Rate (FNMR)
  achieved at a False Match Rate (FMR) of $1.0\%$ or $0.1\%$ respectively.
- Fisher Discriminant Ratio (FDR) (Poh & Bengio, 2004), which quantifies the separability
  of genuine and imposter distributions.

**Recognition experiments.**    In the experiments, we train the CosFace recognition model (Wang
et al., 2018) on the produced synthetic datasets. To determine their suitability, we evalute the per-
formance of the model on five real-world verification benchmarks. These include:

- **Labeled Faces in the Wild (LFW)** Huang et al. (2007), which is an unconstrained web-
  scraped verification dataset of $13,233$ face images of $5749$ identities.

- **Cross-Age Labeled Faces in the Wild (CA-LFW)** Zheng et al. (2017), which is a subset of LFW Huang et al. (2007) with 7156 images of 2996 identities, aimed at evaluating verification performance across a given age gap.

- **Cross-Pose Labeled Faces in the Wild (CP-LFW)** Zheng & Deng (2018), which is a LFW Huang et al. (2007) subset that is suited specifically for evaluating cross-pose verification performance. It includes 5984 face images of 2296 identities captured in various poses.

- **AgeDB-30** Moschoglou et al. (2017), which is a dataset of in-the-wild face images, suited for evaluating verification performance across a 30 year age gap. The dataset comprises 16, 488 images of 568 identities.

- **Celebrities in Frontal-Profile in the Wild (CFP-FP)** Sengupta et al. (2016), which is a verification dataset that is aimed at evaluating cross-pose performance, in particular of frontal and profile poses. In total, it contains 7000 images of 500 identities, each with 10 frontal and 4 profile images.

Summaries of the datasets can also be found in Table 4. Each benchmark is then formed with 3000 genuine and 3000 imposter image pairs of a given verification dataset, with an image resolution of $112 \times 112$. To limit the influence of race and gender, the CA and CP verification pairs are sampled from the same race and gender.