# OpenReview forum: "ID-Booth: Identity-consistent image generation with diffusion models"
_ICLR.cc/2025/Conference — ICLR 2025 Conference Withdrawn Submission_

### Official Review · Reviewer_TqM8 · 2024-10-25

**Soundness:** 2
**Presentation:** 3
**Contribution:** 1
**Rating:** 3
**Confidence:** 5

**Summary:**

ID-Booth proposes a novel way to fine-tune diffusion models specifically for accurate and diverse facial image generation. Existing text-conditioned diffusion models are broadly trained on a vast amount of data, but their training and conditioning methods are not sufficiently granular to facilitate the accurate and consistent generation of specified human faces/identities. Previous works have proposed methods to close this gap but have their own limitations, such as overfitting/limited diversity. ID-Booth builds upon these works by introducing a identity-focused triplet loss meant to improve identity-consistent image generation while retaining the existing image synthesis capability of the diffusion model. They provide qualitative and quantitative experiments to support their claim that ID-Booth facilitates better inter-identity consistency and intra-identity separability/diversity image generation. They also claim that the performance of a a facial recognition model can be improved, both generally and over other identity-specific diffusion methods, by training on facial data synthesized by ID-Booth. They importantly note that this is a useful application with regards to human facial data due to its inherently sensitive nature, difficulty to collect, and ethical, consensual, and privacy concerns.

**Strengths:**

1. The problem regarding human facial data and the authors; discussion surrounding this problem is valid and important.
2. The quantitative metrics used to compare and evaluate their method are insightful and correct.
3. The Related Works section and methodology background are informative and provide sufficient context for the reader.
4. Overall, the writing and clarity of the paper is strong.

**Weaknesses:**

1. The novelty is severely limited. The entire contribution of this paper is summarized in Section 3.4, while the rest of the methodology section is background context from previous papers that are used in the ID-Booth method.


2. The quantitative results are not significant, and even less convincing. With regards to Table 1, ID-Booth does not outperform the two comparative baselines on the image quality metrics (FID and CCMD) in a majority of the settings, across both models (SD-2.1 and SD-XL) and datasets (TFD and FFHQ). In the cases where ID-Booth does perform the best in FID or CCMD, the improvements are negligible, with reported margins ranging from +0.02 to +0.1 over the next-best performing method. The same behavior is seen in the Precision and Recall metrics, where ID-Booth fails to outperform the previous methods across both models and datasets, yet when it does the margin is negligible. When ID-Booth is the best performing model with respect to Precision and Recall, the largest margin of improvement over previous methods is only 0.008. However, the main focus of this paper is the consistency and accuracy of the human facial identities when generating images with ID-Booth, which these previous metrics do not account for, hence the use of the CR-FIQA metric in Table 1. Yet even then ID-Booth is not the best performing method in this metric, which calls into question the contributions of the paper if previous methods already outperform ID-Booth in this facet.

2a. Additional quantitative results are given in Table 2,  where similar behavior to Table 1 is observed - ID-Booth does not outperform previous methods, and when it does it is not by much with only one or two exceptions (FMR100 and FMR1000 for synthetic identities with complex prompts with SD-2.1, and even this is a stretch and is more for transparency on my end). I will note that the FDR metric, which measures the linear separability of two classes, is of specific interest here with respect to facial image intra-class separability. ID-Booth DOES achieve some significant results here, with improvements of up to ~+5.2 seen in some cases, however these alone are not enough to outweigh the aforementioned issues (and the other weaknesses listed below). Moreover, Table 3 also serves as a good measure of the quality, consistency, and accuracy of the ID-Booth generated images through classification. Again, sub-optimal results are reported which occludes what exactly the benefit of ID-Booth is - quantitatively, it is negligible.


3. I understand that image generation cannot only be examined quantitatively and it is important to evaluate the qualitative results as well. However, the only qualitative comparison given is Figure 4 which lacks any significant discussion in the text. The figure itself is confusing as the prompt for each image is not given, which does not describe each method's accuracy to the emotion depicted (assuming one was given in the prompt). Figure 3 is also qualitative, while not in the sense of image quality, however there is close to no difference across the graphs in each setting. I am not sure why it was included as it actually shows that ID-Booth does not separate itself from previous methods.

4. Severe lack of ablations. Many pre-trained models are used in this method, such as ResNet-101 for CR-FIQA and ArcFace for the cosine similarities in Figure 4. This is a smaller note, but these are older models and I wonder if performance would differ when using different or more recent pre-trained models.

In summary, the quantitative and qualitative experiments chosen for the paper are appropriate, but the actual results are very, very poor. Moreover, there is little to no novelty, and the proposed method seems to make no difference in the results.

**Questions:**

I have some questions/comments:

1. Why is the Precision for TFD in Table 1 so low (near-zero)?
2. Why is there only one column for CR-FIQA in Table 1, as opposed to two separate columns for the TFD and FFHQ dataset?
3. Maybe adding some scale for the metrics would help in understanding the significance. For example, is a 0.8 improvement in FDR significant? In the case of a well-known metric like accuracy which is scaled with respect to 100, the answer would be no, but maybe the improvements of ID-Booth could be better highlighted if this was added or discussed.

---

### Official Review · Reviewer_1KSK · 2024-10-30

**Soundness:** 1
**Presentation:** 2
**Contribution:** 1
**Rating:** 3
**Confidence:** 5

**Summary:**

The paper proposes a triplet identity objective designed to finetune a pre-trained diffusion model. This finetuning aims to enhance the generation of face images that are both highly diverse and consistent with the identity. Additionally, the paper conducts a thorough evaluation of the generated images, focusing on aspects such as image quality, identity consistency and diversity, and the recognition performance of models trained using these generated face images.

**Strengths:**

(1) The paper employs a range of quantitative metrics, including CMMD and CRFIQA, to assess the quality of the generated images.
(2) The paper aims to address privacy concerns by generating large-scale training data through the authorization of a small number of images, suggesting an effective method for maintaining privacy.

**Weaknesses:**

（1）The primary distinction between the proposed method and PortraitBooth is the integration of a triplet identity loss function during the fine-tuning stage. While triplet identity loss functions are commonly utilized to enhance model performance in face recognition tasks, the main contribution of this paper is to demonstrate the effectiveness of this specific loss function in fine-tuning the diffusion model. However, based on the results presented in Table 1 and Table 3, there appears to be no significant improvement, which raises questions about the added value of this modification.
（2）The proposed method involves fine-tuning the diffusion model for each individual identity. However, the stated objective of the research is to develop a large-scale face training dataset aimed at mitigating privacy issues, potentially involving over a million identities. A significant concern arises regarding the scalability of the proposed method to accommodate such a vast number of identities. The feasibility of expanding this method to meet the demands of the large-scale dataset described needs to be addressed.
（3）The recognition outcomes presented in Table 3 of the manuscript are not particularly informative. The metric used for evaluation in these datasets is accuracy (acc), and the reported accuracy near 0.5 suggests performance akin to random guessing. Therefore, the minor improvements observed do not convincingly demonstrate the efficacy of the proposed method.
（4） This paper does not include comparisons with previous state-of-the-art (SOTA) methods, such as IDifface mentioned in the related work section.

**Questions:**

The novelty and effectiveness of the method require further explanation. Please refer to the details in the Weakness section.

---

### Official Review · Reviewer_Dry7 · 2024-10-31

**Soundness:** 2
**Presentation:** 3
**Contribution:** 1
**Rating:** 3
**Confidence:** 4

**Summary:**

This paper extends the idea of PortaitBooth by adding a triplet identity loss term to fine-tune large text-to-image models (e.g., SD-XL). This is done to generate identity-preserving diverse images with the goal of introducing a synthetic face recognition dataset without the privacy issues rigged with existing large face recognition datasets.

**Strengths:**

* The writing of the paper is good.
* The authors plan to release their code.

**Weaknesses:**

* The paper's contribution is limited and can be considered just a minor extension to PortraitBooth by adding their triplet identity objective.

* It is not clear what the paper wants to show. In the beginning, the authors raised the issue of GDPR with biometric datasets (i.e., face recognition datasets) and the consent of the people in such datasets. This is because such datasets are usually scraped from the web and don’t have explicit consent. At the same time, the paper builds their method on SD-XL or SD 2.1 which were trained on datasets such as LAION-5B which have the same GDPR issues (apart from objects and scenes they contain millions of face images). When authors use such models to propose a synthetic data generation methodology authors need to show that the fine-tuned model does not contain for example identity presented in the training datasets of the SD-XL, otherwise it still contains the issues that authors trying to solve.

* Although the results of the paper are inconclusive (there is no clear advantage of their additional loss-term ), additionally
the paper uses the InceptionV3 model (probably trained on the ImageNet, not mentioned in the paper) which there are a lot of studies like [1] and [2] that show the inefficacy of this network for evaluation. This was used for reporting the FID, Recall, and Precision. The usage of recent metrics such as MMD is acknowledged but as mentioned there is no clear advantage over DreamBooth or PortraitBooth.

   [1] The role of imagenet classes in frechet inception distance. https://arxiv.org/pdf/2203.06026

   [2] Exposing flaws of generative model evaluation metrics and their unfair treatment of diffusion models
https://proceedings.neurips.cc/paper_files/paper/2023/hash/0bc795afae289ed465a65a3b4b1f4eb7-Abstract-Conference.html

* The paper claims that they got substantial improvement [514-516] over the FR system trained on the original TFD dataset (the dataset they used for fine-tuning the SD-based generator), I would not call the 0.5% to 2.8% on the average of 54% accuracy substantial (which is very low), beside the fact that an ArcFace based pre-trained network used in authors fine-tuning approach. As datasets such as LAION-5B contain millions of face images, we expect to see much more improvement to show the method is valuable (besides the fact that the TFD dataset is small).

**Questions:**

* Please refer to the weaknesses section.

---

### Official Review · Reviewer_CvtJ · 2024-11-05

**Soundness:** 2
**Presentation:** 2
**Contribution:** 1
**Rating:** 3
**Confidence:** 2

**Summary:**

This paper presents a new approach to generating synthetic face datasets for training face recognition (FR) models. The paper proposes a triplet based ID preservation loss to finetune the diffusion model. While the paper makes a strong case for its method, several areas require further clarification and improvement.

**Strengths:**

Strength:
Triplet Identity Training Objective: This is the central contribution. The authors introduce a novel triplet loss that considers both positive (images of the target identity) and negative (images of other identities) examples during training. The approach attempts to avoid overfitting to the training data while still ensuring high-quality synthetic images and adequate inter-identity separability.

**Weaknesses:**

Weakness:
Weak Face recognition Baseline: The FR model trained on synthetic dataset is under-performing previous methods. Arc2Face or DCFace creates synthetic dataset for training FR models and the FR performance can be much higher than Table 3. The low performance needs an explanation.
Speed: To training large scale dataset, the model needs to be able to generating images efficiently. However, this method requires 1. finetuning and 2. subsequently generating which slows down the sampling speed. Speed analysis to show how much portion of time is spent on finetuning is spend would clarify this.

**Questions:**

Addressed in weakness section.

---

### Note · Authors · 2024-11-13

**Comment:**

We sincerely thank the reviewers for their invaluable comments. We will work to improve the paper further and submit it elsewhere.

**Withdrawal Confirmation:**

I have read and agree with the venue's withdrawal policy on behalf of myself and my co-authors.